# The Effect of Abusive Supervision on Safety Behaviour: A Moderated Mediation Model

**DOI:** 10.3390/ijerph182212124

**Published:** 2021-11-18

**Authors:** Xinyong Zhang, Zhenzhen Sun, Zhaoxiang Niu, Yijing Sun, Dawei Wang

**Affiliations:** 1Department of Applied Psychology, Guangdong University of Foreign Studies, Guangzhou 510420, China; zhangxinyong@gdufs.edu.cn; 2School of Psychology, Shandong Normal University, Jinan 250300, China; 2020305004@stu.sdnu.edu.cn (Z.S.); 2020305017@stu.sdnu.edu.cn (Z.N.); 2020305008@stu.sdnu.edu.cn (Y.S.)

**Keywords:** abusive supervision, safety motivation, conscientiousness, safety behavior

## Abstract

Leadership behavior has an impact on the behavior of employees. Previous studies have mainly studied the impact of positive leadership behaviors on employees’ behaviors, but there is an absence of research on the impact of negative leadership behaviours (abusive supervision) on safety behaviours (including safety participation and safety compliance). In this study, 599 front-line employees in the petrochemical industry were selected as subjects. Abusive supervision, safety behaviour, safety motivation and a conscientiousness questionnaire were used as measurements to explore the relationship between abusive supervision and employee safety behaviors, and to further explore the roles of safety motivation, conscientiousness and the relationship between them. This study found that abusive supervision is negatively related to employee safety behaviours (safety compliance and safety participation); that safety motivation plays a mediating role in the relationship between abusive supervision and employees’ safety behavior; and that conscientiousness moderates the role of safety motivation between the relationship of abusive supervision and employees’ safety behaviour. With a higher level of conscientiousness, the indirect relationship between abusive supervision and employee safety behaviours is weaker. Finally, we discuss the theoretical and practical significance of these findings for abusive supervision and the management of safety behaviours.

## 1. Introduction

In recent years, leaders’ management behaviors have received increasing attention, with news of abusive supervision from leaders frequently appearing on social media. For example, online sites have previously reported that in one banking company “a new employee was insulted and slapped by the leader for not drinking alcohol”. Since the beginning of the 21st century, approximately half of the respondents in a survey had said that they had been improperly supervised by a leader [1]. In the United States, the losses caused by abusive supervision are as high as 23.8 billion USD each year [2]. In China, due to its particular organizational culture, abusive supervision is very common. Approximately 70% of employees report that they have experienced non-verbal aggression, such as neglect, indifference, etc., in their workplace [3]. As a result, employees engage in negative organizational behaviours, and their development is hindered [4].

Abusive supervision is considered the dark side of leadership. It is characterized by a supervisor who continues to exhibit hostile verbal and nonverbal behaviours to subordinates (excluding physical contact) [5]. Negative leadership, such as intrusive leadership, can lead to bad work reactions from employees [6]. Therefore, as a kind of leadership behavior similar to abusive supervision that does not respect employees, abusive management may also lead to bad work reactions from employees. Studies have found that abusive supervision can lead to negative psychological states and behaviours of employees, such as psychological distress, compulsory citizenship behaviors, resignation, and a reduction of innovative behaviours. [7,8,9]. In the context of traditional Chinese culture in organizations, the occurrence of abusive supervision is more frequent [10]. Studies have shown that abusive supervision has a negative effect on safety behaviour [11], that is, abusive supervision may lead to low-level safety behaviours. Low-level safety behaviors easily cause safety accidents, while in the petrochemical industry, a high-risk industry, once a safety accident occurs, casualties and economic losses are considerable [12]. Therefore, it is necessary to conduct research on the relationship between abusive supervision and safety behaviours and its mechanisms in the petroleum industry to effectively take measures to improve personal safety behaviour, reduce the incidence of enterprise safety accidents, and improve workplace safety.

The relationship between abusive supervision and safety behaviours can be affected by many factors. Few researchers have explored its inner psychological mechanism. Since leadership style affects safety motivation and safety motivation is closely related to safety behaviours [13,14], safety motivation may play a mediating role between the abusive supervision and the safety behaviors. Studies have shown that different levels of conscientiousness may have a moderating effect on the relationship of individual behaviours [15,16]. Therefore, conscientiousness may also moderate the effect of safety motivation; however, there is no relevant research to clarify this internal relationship mechanism. This study incorporates the variables of safety motivation and conscientiousness into the study of the relationship between abusive supervision and safety behaviours to explore its internal mechanism.

This research makes three contributions to the literature. First, employees of petrochemical companies were used as subjects to explore the relationship between abusive supervision and employee safety behaviour. The result revealed the negative consequences of abusive supervision and provided a direct way to improve the safety behaviors of employees in petrochemical companies. Second, the study explored the mediating mechanism of safety motivation between abusive supervision and employee safety behaviour. This mediation mechanism reflects the important role of safety motivation in the relationship between abusive supervision and employee safety behaviours. It is helpful for researchers to understand the indirect effect of abusive supervision on employee safety behaviours. Third, it further explored the moderating role of conscientiousness in the relationship between safety motivation and employee safety behaviour, which reflects the importance of conscientiousness as a variable on employee safety behaviors and provides a basis for improving employee safety behaviours through the cultivation of an employee’s conscientiousness. The model is shown in Figure 1.

## 2. Theory and Hypothesis

### 2.1. Abusive Supervision and Safety Behaviour

Safe production behaviour refers to employees’ behaviours complying with safe production operating procedures, participation in various safety activities, and the improvement of safe production. In other words, as a separate area of work performance, safety behavior includes two components, safety compliance and safety participation [17]. Safety compliance is generally considered to be the core safety activity of maintaining a safe workplace, such as wearing protective equipment and obeying safety rules. Employees who do not comply with specific safety rules or procedures are considered to be reducing safety compliance or violating rules. Violations may cause accidents, which may cause physical injuries and economic losses [17]. Safety participation includes voluntary safety-related behaviours, such as helping colleagues and volunteering to participate in safety-related activities, which is parallel to the performance of organizational citizenship behaviour. Such voluntary behaviours may not directly promote safety in the workplace, but as an organizational citizenship behaviour in the safety field, they can promote the development of a safe environment [18].

Abusive supervision is characterized by a supervisor who continues to exhibit hostile verbal and nonverbal behaviours to subordinates (excluding physical contact) [5]. Studies have found that employees who have experienced abusive supervision tend to reduce their work performance [19]. Safety behaviours, as a part of their work performance, also decrease after abusive supervision [20]. Experiencing abusive supervision may trigger strong negative emotions and responses from subordinate employees [21], making employees unwilling to comply with the organization’s safety regulations and resulting in a lower level of safety compliance. At the same time, abusive supervision is characterized by long-term abuse, including public ridicule, threats, deliberate concealment of information, and silence [22]. Social exchange theory holds that the process of social interaction is an exchange process. Exchange includes not only material exchange, but also psychological (or social) exchange, such as support, trust, self-esteem and prestige [23]. That is, exchange is accompanied by economic and social expectations (such as treatment and emotional satisfaction) [24]. Abusive supervision that does not meet the social expectations of employees leads to employees’ negative feedback. Employees tend to engage in unsafe behaviours in response to the abusive supervision of leaders and are unwilling to participate in the safety construction of the organization, that is, showing a lower level of safety participation. In summary, this research proposes the following hypotheses:
**H1a.** *Abusive supervision is negatively correlated**with employee safety compliance.*
**H1b.** *Abusive supervision is negatively correlated**with employee safety participation.*

### 2.2. The Mediating Role of Safety Motivation

Safety motivation refers to the willingness of employees to perform work in a safe manner, including their voluntary efforts to complete work tasks safely [25]. Organizational situational factors (leadership style) are related to the stimulation of employees’ safety motivation [17,18] and different leadership styles have different influences on safety motivation. Positive leadership styles can stimulate employees’ safety motivation [26], and negative leadership styles will reduce employees’ safety motivations while increasing occupational safety risks [27]. Abusive supervision, as a negative leadership style, may reduce employees’ safety motivation.

Studies have pointed out that safety motivation has an important influence on employees’ safety behavior [28]. Research has also found that leadership is significantly related to safety motivation and significantly affects safety compliance and safety participation [13]. There is a relationship between leadership and safety-related behaviours (compliance and participation) [29,30]. The principle of equity in social exchange theory means that individuals must conduct internal cost-benefit analyses before participating in a social exchange [31]. When employees experience abusive supervision, they conduct internal cost-benefit analyses and their safety motivation is generated. The lack of support from leaders will make employees choose to reduce their investment in balancing the costs, that is, reduce their own safety motivation and further reduce their compliance with the safety requirements of the organization. Consequently, they function at a lower level of safety compliance. Additionally, social exchange theory holds that human interaction in organizations is essentially a series of exchanges based on the principle of reciprocity [32]. Reciprocity refers to the principle that when one party provides the resource that the other party needs, the other party will reciprocate with the resource that the original party needs while negative reciprocity means that when one party’s resources are damaged, it will retaliate to achieve balance. Therefore, abusive supervision will lead to negative reciprocity, which reduces the safety motivation of employees and reduces their investment in organizational safety construction, thereby showing a lower level of safety participation. In summary, this research proposes these research hypotheses:
**H2a.** *Safety motivation plays a mediating role between abusive supervision and safety compliance.*
**H2b.** *Safety motivation plays a mediating role between abusive supervision and safety participation.*

### 2.3. The Moderating Effect of Conscientiousness

Conscientiousness refers to the characteristic of an individual, including deep thinking, observing norms and rules, and self-discipline. It is the most important personality trait that affects employees’ work behaviors in the five personality studies of Mount and Barrick (1998) [33]. Different levels of conscientiousness may have varied effects on the perception of individual behaviours [15,16]. Regarding the moderating role of conscientiousness, many studies have confirmed that conscientiousness can moderate the relationship between negative emotions and the anti-productive work behaviours of employees [34]; studies also show that conscientiousness moderates the relationship between optimism and work engagement [35]. Therefore, conscientiousness may moderate the relationship between an employee’s internal state and their external individual behaviors (safety motivation and safety behaviour). According to social exchange theory and self-determination theory, when employees experience abusive supervision where their own safety motivation is affected and changed, employees may reduce their safety behaviors. However, due to individual differences in conscientiousness, safety motivation affects employees’ safety behaviours (safety compliance and safety participation) at different levels.

Studies have found that highly responsible employees are more self-disciplined and generally abide by ethical rules and maintain their safety motivation, thus showing better compliance with safety regulations [34]. The behaviours of low-conscientiousness employees depend on the situation, while the behaviours of high-conscientiousness employees are more self-disciplined and depend on personal internal judgement, such as conscientiousness and mission [36]. Therefore, after experiencing abusive supervision, highly responsible employees still feel a sense of mission and conscientiousness. The positive effect of safety motivation on safety behaviour is enhanced, and their safety motivation is strong. Employees then believe that they are obligated to act to maintain the safety of the team, that is, to show a higher level of safe participation. In summary, this research proposes these research hypotheses:
**H3a.** *Conscientiousness moderates the mediating role of safety motivation in abusive supervision and safety compliance.*
**H3b.** *Conscientiousness moderates the mediating role of safety motivation in abusive supervision and safety participation.*

## 3. Methods

### 3.1. Sample and Procedures

Participants were recruited from a domestic petroleum company in China. After explaining the purpose of the study to the person in charge of the enterprise and obtaining consent, we randomly selected 654 employees. These front-line workers, mainly oil miners, are responsible for safety inspections, pipe cleaning and production measurement. If they do not maintain safety, then oil spills, blowouts and other dangerous events may occur, seriously threatening the safety of life and property. A total of 654 questionnaires were sent out, and 599 were recovered, with an effective rate of questionnaire collection of 91.6%. In this study, 63.8% of the participants were male employees, 89.6% were married, 45.4% were aged between 41 and 50, 64.8% had worked for more than 10 years, and 56.9% had a high school education. In addition, we used SPSS 25.0, Process 3.0 (Hayes, Columbus, OH, USA) and Mplus 8.0 (Muthén & Muthén, Los Angeles, CA, USA) statistical analysis software to analyse the data and determine the relationship between the research hypotheses.

All procedures performed in this study involving human participants were conducted in accordance with the ethical standards of the Academic Board of Shandong Normal University and with the 1964 Helsinki Declaration and its later amendments or comparable ethical standards. Informed consent was obtained from the leaders and employees of each company. The information of all the participants was kept strictly confidential, with each participant reserving the right to withdraw from the study at any time.

The data was collected in two stages. The questionnaire presented at stage 1 included demographic information (number, gender, years, marital status, education, work tenures, etc.), as well as the abusive supervision questionnaire and safety motivation questionnaire. Twenty days later, the questionnaires at stage 2 were collected, which included demographic information, the conscientiousness questionnaire and the safety behaviors questionnaire.

### 3.2. Measures

#### 3.2.1. Abusive Supervision Scale

We used the abusive supervision scale developed by Tepper et al. (2000) [5] and translated by Aryee et al. (2007) [37] to measure employees’ perceived abusive supervision. The scale contains 15 items, which are scored based on the subjective feelings of subordinates, such as “My superior laughed at me” and “My superior was angry with me because of something else”. All items were rated on a five-point scale ranging from 1 (disagree) to 5 (agree). The Cronbach’s α coefficient in our study for this scale was 0.980.

#### 3.2.2. Safety Behavior

We measured safety behavior using the safe behavior scale developed by Neal and Griffin (2006) [16] and translated by Ye et al. (2014) [38]. The scale contains two dimensions, safety compliance and safety participation. Safety compliance includes six items, such as “I use all the necessary safety equipment to do my job”. Safety participation includes five items, such as “I promote the safety program within the organization”. All items were rated on a seven-point scale ranging from 1 (Strongly Disagree) to 7 (Strongly Agree). The Cronbach’s α coefficient in our study for this scale was 0.911.

#### 3.2.3. Safety Motivation Scale

We measured safety motivation using the safety motivation scale developed by Neal and Griffin (2006) [16] and translated by Tang and Zheng. (2010) [39]. The scale consists of three items, such as “I feel it is important to be safe at all times”. All items were rated on a five-point scale ranging from 1 (disagree) to 5 (agree). The Cronbach’s α coefficient in our study for this scale was 0.849.

#### 3.2.4. Conscientiousness Scale

We used the conscientiousness scale developed by Roberts et al. [40] and revised by Yang, Wang and Cao (2010) [41] to measure conscientiousness. The scale contains 12 items and 4 dimensions (dedication of duty, solidarity and helping others, diligence, and achievement pursuit). Dedication of duty includes three items, and one of its sample items is “Compliance with professional ethics”. Solidarity and helping others include three items, such as “Maintaining amicable and friendly relationships with colleagues”. Diligence efforts include three items, such as “Actively learning relevant knowledge”. The achievement pursuit consists of three items, such as “The pursuit of excellence”. The scale was measured on a five-point scale ranging from 1 (strongly disagree) to 5 (strongly agree). The Cronbach’s α coefficient in our study for this scale was 0.721.

## 4. Results

### 4.1. Common Method Bias

This study adopted the Harman single-factor technique to estimate the influence of common method bias. The results showed that six factors emerged, with an interpretation rate of the population variance of 71.49%. The interpretation rate of the first common factor was 32.955%, indicating that there was no serious common method bias in this study [42,43].

### 4.2. Confirmatory Factor Analysis

Confirmatory factor analysis was performed using Mplus8.0 (Table 1). With CFA, we investigated the measurement models for several different factors and compared them with the five-factor model. The results showed that the data fitting of the five-factor model was better than that of the other factor models (χ^2^ = 2181.193, df = 750, χ^2^/df = 2.912, RMSEA = 0.056, CFI = 0.935, TLI = 0.929, SRMR = 0.075), that is, the five-factor model was more suitable for data fitting in this study than the other models, indicating that the participants could clearly distinguish among the different factors and that the discriminant validity of this study was good.

### 4.3. Correlation Analysis

Table 2 shows the mean value, standard deviation, and correlation coefficient of the study variables. The results showed that abusive supervision was negatively correlated with safety motivation, safety compliance and safety participation (*t* = −0.273, *p* < 0.01; *t* = −0.230, *p* < 0.01; *t* = −0.213, *p* < 0.01; *t* = −0.201, *p* < 0.01), H1a and H1b were supported; safety motivation was positively correlated with safety behavior, safety compliance and safety participation (*t* = 0.361, *p* < 0.01; *t* = 440, *p* < 0.01; *t* = 0.236, *p* < 0.01); and conscientiousness was positively correlated with abusive supervision, safety compliance and safety participation (*t* = 0.110, *p* < 0.01; *t* = 0.179, *p* < 0.01; *t* = 0.124, *p* < 0.01; *t* = 0.186, *p* < 0.01).

### 4.4. Mediation Analysis

Regression analysis was carried out. As shown in Table 3, after controlling for gender, marital status, years, education and work tenures, the regression coefficient of abusive supervision on safety motivation was significant (*β* = −0.223, *t* = −6.163, *p* < 0.001). When both abusive supervision and safety motivation were included in the regression equation, the regression coefficient of abusive supervision on safety compliance was not significant (*β* = −0.064, *t* = −1.904, *p >* 0.05), indicating that safety motivation played a full mediating role in the relationship between abusive supervision and safety compliance. The regression coefficient of abusive supervision on safety participation was significant (*β* = −0.138, *t* = −3.419, *p* < 0.001), indicating that safety motivation played a partial mediating role in the relationship between abusive supervision and safety participation.

According to Ribeiro et al. (2018) and Lee et al. (2019), this study tested H2a and H2b through two regression models using the PROCESS macro (Model 4) provided by Hayes (2013). The variables in the analysis were mean centred with 95% CIs. Specifically, the direct effect of abusive supervision on safety compliance was not significant, and the direct effect was −0.064, 95% CI [−0.131, 0.002]. The indirect effect of abusive supervision on safety compliance through safety motivation was significant, and the indirect effect was −0.107, 95% CI [−0.148, −0.067]. The direct effect of abusive supervision on safety participation was significant, and the direct effect was −0.138, 95% CI [−0.217, −0.059]. The indirect effect of abusive supervision on safety participation through safety motivation was significant, and the indirect effect was −0.052, 95% CI [−0.081, −0.029]. Thus, H2a and H2b were supported.

### 4.5. Moderated Mediating Analysis

According to Ribeiro et al. (2018) and Lee et al. (2019), this study tested H3a and H3b through two regression models using the PROCESS macro (Model 14) provided by Hayes (2013). The variables in the analysis were mean centred with 95% CIs. The bootstrap estimate was based on 5000 bootstrap samples. As shown in Table 3, the interaction between abusive supervision and conscientiousness was significant (*β* = 0.108, *t* = 2.916, *p* < 0.01; *β* = 0.114, *t* = 3.411, *p* < 0.001), and as shown in Table 4, the three indices of moderated mediation did not include zero (moderated mediation index1 = −0.024, 95% CI [−0.048, −0.001]; moderated mediation index1 = −0.022, 95% CI [−0.041, −0.002]). Therefore, H3a, H3b were supported.

Specifically, under the condition of low conscientiousness, the effect of abusive supervision on safety compliance through safety motivation was significant (indirect effect = −0.069, *SE* = 0.017, 95% CI [−0.106, −0.039]). Under medium conscientiousness, abusive supervision had a significant effect on safety compliance through safety motivation (indirect effect = −0.091, *SE* = 0.018, 95% CI [−0.128, −0.057]). Under high conscientiousness, abusive supervision had an obvious effect on safety compliance through safety motivation (indirect effect = −0.112, *SE* = 0.023, 95% CI [−0.158, −0.067]).

Under the condition of low conscientiousness, abusive supervision had a significant effect on safety participation through safety motivation (indirect effect = −0.022, *SE* = 0.016, 95% CI [−0.056, 0.007]). Under medium conscientiousness, abusive supervision had a significant effect on safety participation through safety motivation (indirect effect = −0.046, *SE* = 0.012, 95% CI [−0.072, −0.024]). Under high conscientiousness, the effect of abusive supervision on safety participation through safety motivation was not significant (indirect effect = −0.069, *SE* = 0.018, 95% CI [−0.106, −0.037]).

To visualize the conditional effect of the different levels of conscientiousness (low, high) on the relationship between safety motivation and safety compliance and safety participation, the interaction effects are shown in Figure 2 and Figure 3.

## 5. Discussion

### 5.1. Theoretical Contribution

#### 5.1.1. Abusive Supervision and Safety Behaviors

Studies have shown that abusive supervision behaviours have a negative relationship with employees’ safety behaviors (including safety compliance and safety participation) [11]. Hypothesis 1a and Hypothesis 1b are supported in this study. The results suggest that disrespecting an employee’s leadership style does lead to poor work behaviours [6]. The reason why employees in the petrochemical industry choose to lower their safety behaviour level after experiencing abusive supervision may be due to the phenomenon of social exchange. According to the principle of negative reciprocity in social exchange theory, leaders may conduct abusive supervision on employees, which then makes the employees realize that they receive worthless feedback from leaders for their work. Based on the principle of negative reciprocity, the employees then regard the safety of the organization with this negative view which results in a lower level of safety compliance. Previous studies have pointed out that the relationship between Chinese managers and employees is characterized by a “high power distance” [44], and traditional Chinese culture emphasizes the idea of peace and harmony as most valuable [45,46]. Therefore, employees will choose alternative retaliatory actions under abusive supervision. That is, they will negatively respect organizational safety, be unwilling to participate in organizational safety construction, and show a lower level of safety participation.

#### 5.1.2. The Mediating Role of Safety Motivation

The research found that safety motivation plays a mediating role in the relationship between abusive supervision and employee safety behaviours (safety compliance and safety participation). Therefore, Hypothesis 2 is supported.

Previous studies have found that leadership style can influence how employees incorporate various company values, such as safety, work motivation, attitude, and work performance [47]. This study also supports this. This study found that when leaders conduct abusive supervision on employees, employees’ motivation for work is damaged, leading to their weaker motivation to perform safe behaviours. According to the theory of social exchange, when employees experience abusive supervision, they will take measures to respond similarly to the abusive supervisor. This is based on the principles of fairness and reciprocity in social exchange theory [32]. Therefore, after experiencing abusive supervision, employees will adopt negative behaviours or reduce their own positive behaviours, that is, reduce their safety motivation.

Studies have found that there is a positive correlation between safety motivation and safety behaviors [48], which is consistent with the results of this research. The reason for this might be that an individual’s behaviour is driven by their motivation. Motivation is more helpful in stimulating desired behaviours and is associated with many positive results, such as promoting employees’ job performances [49,50]. Therefore, when employees have a higher level of safety motivation, they will be effectively motivated to comply with the organization’s safety regulations, show a higher level of safety compliance, and participate in safety construction.

Abusive supervision reduces the safety motivation of employees, which further reduces employee safety behaviours (safety compliance and safety participation). The reason why abusive supervision leads to the declination of safety motivation and employee safety behaviours may be that while personalized care and a supportive leadership style of leaders can positively affect safety motivation [51,52], abusive supervision does not provide personalized care and support and is negatively correlated with supportive feelings such as belonging [11]. Therefore, it negatively affects safety motivation and further leads to a reduction in the level of employee safety behaviour. In addition, this study found heterogeneity in the mediating mechanism of safety motivation in the relationship between different safety behaviours and abusive supervision. Safety motivation plays a complete mediating role in the relationship between abusive supervision and safety compliance and plays a partial mediating role in the relationship between abusive supervision and safety participation. These results imply that the influence of abusive supervision on different safety behaviours of employees is different depending on safety motivation. This may be because safety compliance is considered in-role behavior, which is a formal behaviour that is included in job requirements, and in-role behavior is inseparable from the key role of individual internal factors (such as motivation, etc.) [53]. Therefore, safety motivation can completely mediate the relationship between abusive management and safety compliance; however, safety participation is an extra-role behavior, which is a voluntary behaviour required by informal regulations and extra-role behaviors not only require individuals to have motivation, but also requires individuals to identify with the organization to continue to appear [54]. Therefore, safety motivation only partially mediates the relationship between abusive management and safety participation.

These findings basically reveal the transmissive role of safety motivation. Abusive supervision will influence safety motivation and then affect employee safety behaviours (including safety compliance and safety participation), thereby encouraging research on the relationship between safety motivation and safety behaviour in the petrochemical industry.

#### 5.1.3. The Moderating Effect of Conscientiousness

The results indicate that conscientiousness plays a moderating role in the relationship between safety motivation and safety behaviours (including safety compliance and safety participation). In addition, the research also shows that the positive promotion effect of safety motivation on safety behaviours has been strengthened for individuals with a high level of conscientiousness. Hypothesis H3a and H3b are supported. This result may be due to a conscientiousness that drives individuals to actively pay attention to and complete their tasks and responsibilities [55,56]. Employees with high levels of conscientiousness are better able to monitor and command their behaviors, and can monitor themselves to take action when motivated. Therefore, conscientiousness enhances the impact of safety motivation on safety behaviours. Employees with a high conscientiousness level will feel driven to achieve their own tasks, achieve a higher level of safety motivation, be willing to take up their own safety responsibilities and participate in the organization’s safety construction.

In summary, this study uses conscientiousness as a moderating variable to study safety behaviours (including safety compliance and safety participation) and to explore the effect of conscientiousness on the mediating role of safety motivation between abusive supervision and employee safety behaviours (including safety compliance and safety participation). This study provides a new perspective for future study.

### 5.2. Practical Contributions

First, this research reveals the negative correlation between abusive supervision and employees’ safety behaviour. According to our research, abusive supervision from a leader can significantly reduce the safety behaviour of employees, so the management style of the leader is particularly important. China is a country with a ‘high power distance’, and companies should take corresponding measures to reduce abusive supervision from leaders. Second, a comprehensive zero-tolerance policy should be adopted for abusive supervision of leaders, and employees should be aware of this policy [57]. Finally, abusive supervision may be inevitable in certain situations, and organizations should establish anti-bullying policies and management systems to deal with abusive supervision, such as improving supervision and anonymous reporting methods [58].

Second, this study found a mediating effect of safety motivation on abusive supervision and employee safety behaviours. Therefore, cultivating employees’ safety motivation may be a key factor for employees to maintain high safety behaviours despite abusive supervision. In the job training process, companies can adopt various methods, such as implementing person–job fit [59], playing safety behavior dramas and safety films, and introducing real cases of safety accidents to enhance employees’ safety motivations and to listen to their safety concerns [60]. In addition, companies can also promote safety messages so that employees can realize the importance of standardized operations and improve their independent safety motivation.

Finally, this study also found that conscientiousness can moderate the mediating role of safety motivation between abusive supervision and employee safety behaviour. This inspires leaders to pay attention to selecting candidates with a strong conscientiousness when recruiting employees. Responsible employees are more likely to abide by safety rules and regulations to avoid safety accidents. For employees with a low conscientiousness or low safety motivation, companies should also implement employee assistance programs (EAP) to help them relieve psychological pressure and build up a positive attitude to work [61]. Companies should promote organizational culture as well as teamwork and conscientiousness construction to increase employees’ conscientiousness, reduce their experiences of abusive supervision, and improve safety motivation and safety behaviours.

## 6. Limitations and Future Direction

This study validated our hypothesis despite some limitations. First, this study was a cross-sectional study, which only provides evidence instead of inference for causality. Therefore, future research can consider adopting empirical sampling to conduct longitudinal research to explore the dynamic changes and causal relationships between abusive supervision and employee safety behaviours. Second, the data collection method of this study relies on the subjective evaluation of the subjects’ self-reports, which may contain certain discrepancies. Future research may consider collecting data from multiple subjects, for example, in future studies, researchers can collect data by matching managers with employees. Finally, this research investigates the relationship between leaders’ negative management styles and employee safety behaviours. Future research may focus on the relationships and mechanisms between other organizational factors and safety behaviours, such as safety climate, organizational safety culture and voice behaviours related to safety.

## 7. Conclusions

There is a significant negative correlation between abusive supervision and employee safety behaviours (safety compliance and safety participation); safety motivation plays a mediating role in the relationship between abusive supervision and employee safety behaviours (safety compliance and safety participation), and conscientiousness moderates the role of safety motivation between abusive supervision and safety behaviours (safety compliance and safety participation). For individuals with a high conscientiousness, safety motivation’s positive role in promoting safety behaviours (safety compliance and safety participation) will be enhanced.

## Figures and Tables

**Figure 1 ijerph-18-12124-f001:**
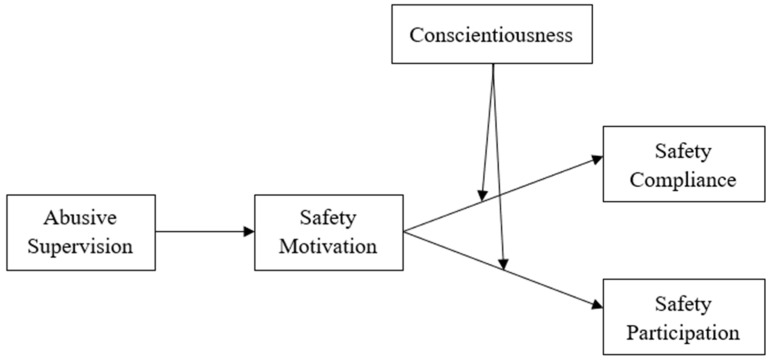
Moderated mediation model.

**Figure 2 ijerph-18-12124-f002:**
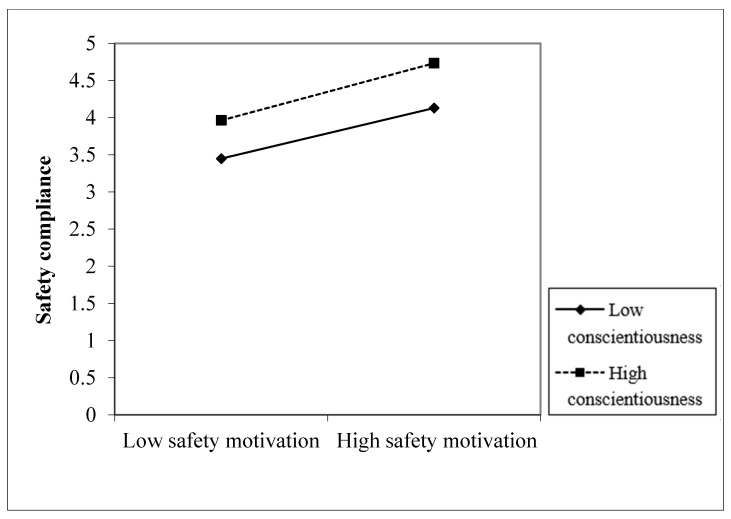
Interaction between safety motivation and conscientiousness on safety compliance.

**Figure 3 ijerph-18-12124-f003:**
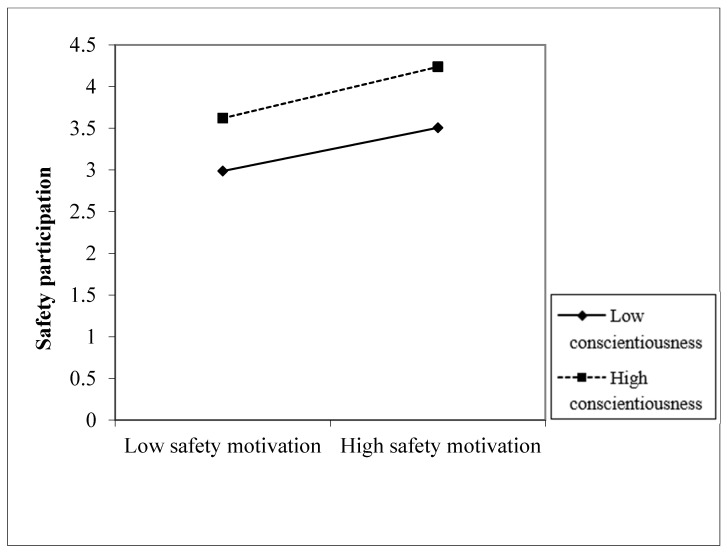
Interaction between safety motivation and conscientiousness on safety participation.

**Table 1 ijerph-18-12124-t001:** Results of confirmatory factor analysis of the measurement models.

Measurement Models	χ^2^	df	χ^2^/df	RMSEA	CFI	TLI	SRMR
Five-factor	2181.193	750	2.912	0.056	0.935	0.929	0.075
Three-factor	6436.539	776	8.295	0.110	0.743	0.728	0.102
Two-factor	7328.914	778	9.420	0.119	0.703	0.686	0.119
One-factor	11184.340	779	15.223	0.149	0.528	0.503	0.179

Note: five-factor, hypothesis model; three-factor, conscientiousness, safety compliance and safety participant combined into one factor; two-factor, abusive supervision and safety motivation combined into one factor, conscientiousness, safety compliance and safety participant combined into one factor; one-factor, five variables combined into one factor. RMSEA: root-mean-square error of approximation; SRMR: standardized root-mean-square residual; CFI: comparative fit index; TLI: Tucker-Lewis’s index.

**Table 2 ijerph-18-12124-t002:** Descriptive statistics and correlations among study variables.

Variable	*M*	*SD*	1	2	3	4	5	6	7	8	9
1. Abusive supervision	2.005	1.008	-								
2. Safety motivation	4.628	0.471	−0.273 **	-							
3. Safety compliance	6.505	0.622	−0.213 **	0.440 **	-						
4. Safety participation	6.019	1.007	−0.201 **	0.236 **	0.595 **	-					
5. Conscientiousness	3.688	0.481	0.110 **	0.047	0.124 **	0.186 **	-				
6. Marital status	1.130	0.397	0.122 **	0.017	−0.030	−0.024	0.049	-			
7. Education	2.570	0.775	−0.152 **	0.121 **	−0.028	0.038	−0.069	0.022	-		
8. Years	-	-	0.034	0.018	0.003	−0.062	−0.086 *	−0.161 **	−0.460 **	-	
9. Work tenures	4.350	0.975	0.021	0.022	0.006	−0.077	−0.048	−0.210 **	−0.449 **	0.743 **	-

Note: ** *p* < 0.01, * *p* < 0.05.

**Table 3 ijerph-18-12124-t003:** Hierarchical regression results.

	Outcome Safety Motivation			Outcome Safety Compliance			Outcome Safety Participation			Outcome Safety Compliance			Outcome Safety Participation		
Variable	*β*	*SE*	*t*	*β*	*SE*	*t*	*β*	*SE*	*t*	*β*	*SE*	*t*	*β*	*SE*	*t*
Gender	0.039	0.036	1.082	0.129	0.032	3.997 ***	0.006	0.039	0.162	0.136	0.032	4.287 ***	0.018	0.038	0.480
Marital status	0.060	0.035	1.692	−0.029	0.032	−0.897	−0.030	0.038	−0.792	−0.037	0.032	−1.168	−0.040	0.037	−1.073
Education	0.115	0.040	2.880 **	−0.111	0.036	−3.071	−0.059	0.043	−1.357	−0.096	0.036	−2.676	−0.033	0.043	−0.771
Years	0.042	0.053	0.791	−0.045	0.048	−0.948	−0.022	0.057	−0.394	−0.023	0.047	−0.486	0.013	0.056	0.226
Work tenures	0.052	0.053	0.996	−0.040	0.048	−0.842	−0.090	0.057	−1.593	−0.047	0.047	−1.001	−0.098	0.055	−1.780
Abusive supervision	−0.223	0.036	−6.163 ***	−0.064	0.034	−1.904	−0.138	0.040	−3.419 ***	−0.080	0.033	−2.390 *	−0.163	0.040	−4.114 ***
Safety motivation				0.414	0.037	11.154 ***	0.221	0.044	4.999 ***	0.405	0.037	11.036 ***	0.204	0.043	4.698 ***
Conscientiousness										0.124	0.033	3.791 ***	0.209	0.039	5.401
Safety motivation×Conscientiousness										0.099	0.031	3.155 **	0.108	0.037	2.916 **
*R^2^*	0.093			0.237			0.086			0.262			0.134		
*F*	10.1642 ***			26.159 ***			7.932 ***			23.226 ***			10.108 ***		

Note: *** *p* < 0.01, ** *p* < 0.01, * *p* < 0.05.

**Table 4 ijerph-18-12124-t004:** Moderated mediation results for safety motivation across levels of conscientiousness on safety compliance and safety participation.

Outcome Variable		Effect Index	*SE*	LLCI	ULCI
Conditional indirect effect at conscientiousness = *M* ± 1*SD*					
Safety compliance	*M* − 1*SD*	−0.069	0.017	−0.106	−0.039
	*M*	−0.091	0.018	−0.128	−0.057
	*M +* 1*SD*	−0.112	0.023	−0.158	−0.067
Safety participation	*M* − 1*SD*	−0.022	0.016	−0.056	0.007
	*M*	−0.046	0.012	−0.072	−0.024
	*M +* 1*SD*	−0.069	0.018	−0.106	−0.037
Index of moderated mediation					
Safety compliance		−0.024	0.012	−0.048	−0.001
Safety participation		−0.022	0.010	−0.041	−0.002

## Data Availability

The correspondence consent is required to use the data.

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
