# Peer review of "The Effect of Abusive Supervision on Safety Behaviour: A Moderated Mediation Model"

_ijerph, 2021, doi:10.3390/ijerph182212124_

Round 1

Reviewer 1 Report

Dear authors,

I think the manuscript has now significantly improved. Only minor comments to fix:

Results section: the sentence "the results showed that there were 6 factors emerged" is incorrrect. It should say "the results showed that 6 factors emerged"

Section 4.2. There is still no explanation as to why you check four factor solutions (five-factor, three factor, two factor, and one-factor). The table is fine, but you need a sentence explaining why you test these four structures.

When you comment results from table 4 you state "Therefore, H3a, H3b were be supported", it should state "were supported".

THe new section in page 17 where you discuss employees with high level of conscientiousness there is a wrong expression "can monitor yourself". It should say "can monitor themselves"

Good luck and congratulations on your work

Reviewer 2 Report

The authors have sufficiently responded to the minor comments I provided in the earlier round. No further need for improvements.

Author Response

Thank you very much!

Reviewer 3 Report

in general the research was well-done and the writing is good.

Table 2 - correlation requires continuous variables. Gender is a dichotomous varilable. Therefore, gender should not be included in the correlation.  Dichotomous variables can be used in regression as dummy variables in which a weight/value is ascribed to a single trait. 

minor writing issues that can be corrected with detailed proofreading:

lines 23-25: "In recent years, the issue of leaders' management behaviors has received increasing attention, with news of abusive supervision by leaders has frequently appeared on social media." (check word use and verb consistency) 

Author Response

This manuscript is a resubmission of an earlier submission. The following is a list of the peer review reports and author responses from that submission.

Round 1

Reviewer 1 Report

I would like to thank the editor of IJERPH for providing me a change to review this article manuscript on the effects on abusive supervision on safety behaviour in a petrochemical industry context.

After carefully reading through the manuscript, I want to express that the manuscript is well written entity with very little to be improved. To me, the literature section seems adequate with good references and the study setting is understandable as is also the analysis part. I can only point some very minor issues, like a) the beginning with a bit odd reference to a social media source.. should that be referenced? b) some references maybe missing; e.g. Deci and Ryan (line 107) and  Roberts et al. (line 224). Then I wouldn't personally use references at all in the results section, but I guess that they can stay..

So, at the end no major issues, only a couple of minor issues that you should consider. Good luck with finalizing the document.

Author Response

Thank you for taking time to provide this in-depth review of our paper. This valuable information has allowed us to improve the manuscript. Below, we clarify the changes that have been made.

1.The beginning with a bit odd reference to a social media source. should that be referenced?

Response: Thank you for your kind advice. The use of social media here is to express that "news about leaders' abusive management frequently appears on social media". In order to make this meaning clearer, we have modified it. (See page1)

In recent years, the issue of leaders' management behaviors has received increasing attention, with news of abusive supervision by leaders has frequently appeared on social media. For example, online sites reported that in a bank company in which “a new employee was insulted and slapped by the leader for not drinking alcohol”.

  1. Some references maybe missing; e.g., Deci and Ryan (line 107) and Roberts et al. (line 224).

Response: Thank you for your kind advice. The references of Roberts et al. (line 224) has been added. Since the discussion of the document " Deci and Ryan " has been replaced, we did not cite the content of this document, so we finally deleted this reference. (See page 21)

Roberts, B. W., Chernyshenko, O. S., Stark, S., & Goldberg, L. R. (2005). The structure of conscientiousness: an empirical investigation based on seven major personality questionnaires. Personnel Psychology, 58(1), 103–139.

Reviewer 2 Report

This manuscript is on an interesting topic about the relationship between abusive leadership and safety behavior as mediated by motivation and moderated by conscientiousness. It is a relevant and interesting topic, and the sample size and methods seem correct, however there is a lot of work needed in this paper to improve the writing of the theoretical framework and elaborate the discussion with richer inputs.

Overall, Social Exchange theory has a main role in this paper. It deserves therefore a better explanation early in the paper. The role of Deci and Ryan’s theory is under-exploited, and not really well connected to the variable choice and arguments provided so far, with an emphasis on different types of motivation which are not properly explained. The authors need to work on a better presentation of the theoretical rationale for the hypotheses, and explain in the discussion how the study contributes to theoretical development and advancement. The discussion is pretty much a repetition of earlier info instead of a rich discussion. Finally, the Study context should be described with regards to the type of jobs and safety implications for this particular sample of employees.

Having said that I offer some suggestions on how to improve the paper, which I hope are useful to the authors.

The introduction should be revised. There is no reference for this incident in a bank, but I googled it and it says the new hire was insulted by colleagues and slapped by a manager (not his manager). Please explain what “cold violence” means. In line 38 the authors mention “mandatory organizational citizenship behavior (OCB)”, please define this concept as traditionally OCB is defined as a discretionary behavior, not mandatory.

Line 60. The contributions of the paper should be elaborated not repeat the main links in the research model.

Line 96. The contribution of social exchange theory is interesting, but should be better described and elaborated.

Section 2.2. The description of different types of motivation according to self-determination theory is confusing (external, controlled, internal external pressure, autonomous…), and most likely not adequate in the context of this study since the authors do not differentiate empirically between different types of motivation. The authors need to either explain the role of autonomous motivation more clearly upfront, or directly refer to safety motivation and its antecedents. The principles of fairness and reciprocity are mentioned here, but not explained til the discussion section, I suggest to explain them here and elaborate on the idea in the discussion.

Section 2.3. the sentence “Different levels of conscientiousness may have a moderating effect on the PERCEPTION OF INDIVIDUAL ATTITUDES AND BEHAVIOURS” does not make sense to me. It affects the perceptions or it affects the relationship between attitudes and behaviours? Please clarify this sentence and what the references state with regards to this issue.

METHODS. Please describe the job of the sample individuals. In the abstract you mention front-line employees, what is their role? What type of safety behaviors are required in their job? What kind of accidents may happen if they do not keep safe? Without this information of the Study context and the sample jobs the practical relevance of your study is not understood. Later on in section 5.1.1. you mention pilot community. It’s confusing.

 The procedure for contacting and obtaining the data is not described. Lines 195 should state “effective response rate”. Please add the initial statement for the items in the Conscientiousness scale.

RESULTS section (an S is missing from the title.  Point 4.2. why do you test all these factor models? A comparison between a five and four-factor models, and maybe a one-factor model should be enough. Table 2: please clarify the difference between years and work time. Please revise the mentions to safety participation and conscientiousness in tables 2 and 3. There is no explanation of the data analysis procedure in the method section, therefore it comes as a surprise that the results are presented for both Safety behavior and the two safety behavior subscales. Figures 2 and 3 are much like figure 1, they can actually be deleted, and a comment added to explain that the three figures are similar.

Overall, the results seem to be accurately described. However, the order in which the results are presented is difficult to follow, in particular section 4.4.

DISCUSSION. A lot new information has been added to the discussion which should have first been introduced in the theory section. For instance, fairness theory, reciprocity, high power distance… These concepts should be introduced earlier on, and then discussed in the discussion with reference and comparison with similar empirical studies.

In my view, the elaboration on the different types of motivation according to Ryan and Deci in the introduction and in the discussion is inappropriate as the study has not empirically distinguished or measured autonomous vs. controlled motivation.

The results on full vs. partial mediation of motivation for the different safety behaviors should be further elaborated and discussed. It is confusing to mix results on both “overall safety behaviors”, and on “specific safety behaviors”, I suggest starting with an overall comment on safety behavior, but discussing results only on specific safety behaviors (compliance and participation).

The authors should elaborate and provide richer inputs in the discussion, at the moment it is pretty much a repetition of the introduction and results.

Formal aspects: References are not in alphabetical order. Punctuation signs need to be reviewed. English needs to be reviewed by a native-level speaker.

Author Response

Thank you for taking time to provide this in-depth review of our paper. This valuable information has allowed us to improve the manuscript. Below, we clarify the changes that have been made.

  1. The introduction should be revised. There is no reference for this incident in a bank, but I googled it and it says the new hire was insulted by colleagues and slapped by a manager (not his manager).
  2. Please explain what “cold violence” means.

Response: Thank you for your kind advice. Abusive management alias “workplace cold violence”, because it is characterized by a supervisor who continues to exhibit hostile verbal and nonverbal behaviours to subordinates (excluding physical contact) [5].

  1. In line 38 the authors mention “mandatory organizational citizenship behavior (OCB)”, please define this concept as traditionally OCB is defined as a discretionary behavior, not mandatory. (See page2)

Response: Thank you for your kind advice. This is a translation error.

The “mandatory organizational citizenship behavior” should actually be called “compulsory citizenship behavior”. CCB refers to personal participation in extra-role activities that always go against one’s will, displaying a distinct dynamic different from voluntary beneficence (Vigoda-Gadot, 2007).

Vigoda-Gadot, E. (2007). Redrawing the boundaries of OCB? An empirical examination of compulsory extra-role behavior in the workplace. Journal of Business and Psychology, 21(3), 377–405.

2.Line 60. The contributions of the paper should be elaborated not repeat the main links in the research model.

Response: Thank you for your kind advice. We have refined the contributions of the paper. (See page 2-3)

This research makes three contributions to the literature. First, employees of petrochemical companies were used as subjects to explore the relationship between abusive supervision and employee safety behaviour. The result reveals the negative consequences of abusive supervision and provide a direct way to improve the safety behaviour of employees in petrochemical companies. Second, the study explored the mediating mechanism of safety motivation between abusive supervision and employee safety behaviour. This mediating mechanism reflects the important role of safety motivation between leadership style and employee safety behaviour, and helps researchers to understand an indirect path such as abusive supervision that affects employee safety behaviour. Third, it further explored the moderating role of conscientiousness in the relationship between abusive supervision and employee safety, which reflects the importance of conscientiousness as a variable on employee safety behaviour, and provides a basis for improving employee safety behaviour through the cultivation of employee conscientiousness. The model is shown in Figure 1.

3.Line 96. The contribution of social exchange theory is interesting, but should be better described and elaborated.

Response: Thank you for your kind advice. We have added the elaboration of social exchange theory. (See page 4)

Social exchange theory holds that the process of social interaction is an exchange process. Exchange includes not only material exchange, but also psychological (or social) exchange, such as support, trust, self-esteem and prestige [22]. That is, exchange is accompanied by economic and social expectations (such as treatment and emotional satisfaction) [23].

4.Section 2.2.

  1. The description of different types of motivation according to self-determination theory is confusing (external, controlled, internal external pressure, autonomous…), and most likely not adequate in the context of this study since the authors do not differentiate empirically between different types of motivation. The authors need to either explain the role of autonomous motivation more clearly upfront, or directly refer to safety motivation and its antecedents.

Response: Thank you for your kind advice. We modified the autonomous motive part. (See page 4)

Safety motivation refers to the willingness of employees to perform work in a safe manner, including their voluntary efforts to complete work tasks safely [24]. Organizational situational factors (leadership style) are related to the stimulation of employee safety motivation [16,17]. Different leadership styles have different influences on safety motivation. Positive leadership styles can stimulate employees' safety motivation [25], and negative leadership styles will reduce employees’ safety motivations and increase occupational safety risks [26]. Abusive supervision, as a negative leadership style, may reduce employee safety motivation.     

  1. The principles of fairness and reciprocity are mentioned here, but not explained til the discussion section, I suggest to explain them here and elaborate on the idea in the discussion.

Response: Thank you for your kind advice. We have adjusted our interpretation about principles of fairness and reciprocity. (See page 4-5)

The principle of equity in social exchange theory means that individuals must conduct internal cost-benefit analyses before participating in social exchange [30]. When employees experience abusive supervision, they conduct internal cost-benefit analyses when their safety motivation is generated. The lack of support from leaders will make employees choose to reduce their investment in balancing the costs, that is, reduce their own safety motivation and further reduce the safety requirements for the organization. Consequently, they function at a lower level of safety compliance; meanwhile, social exchange theory holds that human interaction in organizational life is essentially a series of exchanges based on the principle of reciprocity [31]. Reciprocity refers to the principle that when one party provides the resource that the other party needs, the other party will reciprocate with the resource that the other party needs. And negative reciprocity means that when one party's resources are damaged, it will retaliate to achieve balance. Abusive supervision will lead to negative reciprocity, which reduces the safety motivation of employees and reduces their investment in organizational safety construction, thereby showing a lower level of safety participation.

  1. The procedure for contacting and obtaining the data is not described.

Response: Thank you for your kind advice. We described the steps for connecting and obtaining data. (See page 7)

The data was collected in two stages. The questionnaire presented at stage 1 included demographic information (number, gender, years, marital status, education, work tenures, etc.), as well as the abusive supervision questionnaire and safety motivation questionnaire. Twenty days later, the questionnaires at stage 2 were collected, which included demographic information, conscientiousness questionnaire and the safety behaviours questionnaire.  

  1. Lines 195 should state “effective response rate”.

Response: Thank you for your kind advice. "Effective response rate" refers to the ratio of questionnaires we send out to valid questionnaires we receive. We changed it to "the efficiency of questionnaire collection" (See page 6)

A total of 654 questionnaires were sent out, and 599 were recovered, and the effective rate of questionnaire collection is 91.6%.

  1. Please add the initial statement for the items in the Conscientiousness scale.

Response: Thank you for your kind advice. We added the initial statement for the items in the Conscientiousness scale. (See page 8)

We used the conscientiousness scale developed by Roberts et al. [38]and revised by Yang, Wang and Cao (2010) [39]to measure conscientiousness. The scale contains 12 items and 4 dimensions (dedication of duty, solidarity and helping others, diligence and achievement pursuit). Dedication of duty includes three items, such as "compliance with professional ethics". Solidarity and helping others include three items, such as "Maintaining amicable and friendly relationships with colleagues". Diligence efforts include three items, such as "actively learning relevant knowledge". The achievement pursuit consists of three items, such as "the pursuit of excellence". The scale was measured on a five-point scale ranging from 1 "strongly disagree" to 5 "strongly agree". The Cronbach’s α coefficient in our study for this scale was 0.721.

5.Section 2.3. the sentence “Different levels of conscientiousness may have a moderating effect on the PERCEPTION OF INDIVIDUAL ATTITUDES AND BEHAVIOURS” does not make sense to me. It affects the perceptions or it affects the relationship between attitudes and behaviors? Please clarify this sentence and what the references state with regards to this issue.

Response: Thank you for your kind advice. It has been suggested in the references that conscientiousness may regulate the relationship between emotion and behavior, so we hypothesize that conscientiousness may regulate the relationship between attitudes and behaviors. And we have modified it. (See page 2)

Different conscientiousness may have a moderating effect on the relationship of individual attitudes and behaviors.

  1. METHODS.
  2. Please describe the job of the sample individuals. In the abstract you mention front-line employees, what is their role? What type of safety behaviors are required in their job? What kind of accidents may happen if they do not keep safe? Without this information of the Study context and the sample jobs the practical relevance of your study is not understood.

Response: Thank you for your kind advice. In the methods section, we added a description of the sample individual's work. (See page 6)

These front-line workers, mainly oil miners, are responsible for safety inspections, pipe cleaning and production measurement. If they do not keep safe, then oil spills, blowouts and other dangerous events may occur, seriously threatening the safety of life and property.

  1. Later on, in section 5.1.1. you mention pilot community. It’s confusing.

Response: Thank you for your kind advice. We deleted the “pilot community”.

  1. Overall, the results seem to be accurately described. However, the order in which the results are presented is difficult to follow, in particular section 4.4.

Response: Thank you for your kind advice. We have added a description of the analytical method. (See page 11-12)

Regression analysis was carried out. As shown in Table 3, after controlling for gender, marital status, age, education level and working years, the regression coefficient of abusive supervision on safety motivation was significant (β = -0.223, t = -6.163, p < 0.001). When both abusive supervision and safety motivation were included in the regression equation, the regression coefficient of abusive supervision on safety compliance was not significant (β = -0.064, t = -1.904, p > 0.05), indicating that safety motivation played a full mediating role in the relationship between abusive supervision and safety compliance. The regression coefficient of abusive supervision on safety participation was significant (β = -0.138, t = -3.419, p < 0.001), indicating that safety motivation played a partial mediating role in the relationship between abusive supervision and safety participation.

According to the bootstrap method proposed by Preacher & Hayes (2004) and Hayes (2013), the mediating effect test was performed by setting 5000 repeated samples and calculating a 95% confidence interval. Specifically, the direct effect of abusive supervision on safety compliance was not significant, and the direct effect was -0.064, 95% CI [-0.131, 0.002]. The indirect effect of abusive supervision on safety compliance through safety motivation was significant, and the indirect effect was -0.107, 95% CI [-0.148, -0.067]. The direct effect of abusive supervision on safety participation was significant, and the direct effect was -0.138, 95% CI [-0.217, -0.059]. The indirect effect of abusive supervision on safety participation through safety motivation was significant, and the indirect effect was -0.052, 95% CI [-0.081, -0.029]. Thus, H2a and H2b were supported.

  1. RESULTS section
  2. An S is missing from the title.

Response: Thank you for your kind advice. The missing "s" has been added.

  1. Point 4.2. why do you test all these factor models? A comparison between a five and four-factor models, and maybe a one-factor model should be enough.

Response: Thank you for your kind advice. The four-factor model takes safety behavior as a factor, and the five-factor model divides safety behavior into safety compliance and safety participation. In this study, because we discussed safety compliance and safety participation separately, the four-factor model was deleted and only the five-factor model was retained.

  1. Table 2: please clarify the difference between years and work time.

Response: Thank you for your kind advice. "Years" refers to an individual's age grades and "Work time" refers to the number of Years an individual has worked. (See page 10,12)

To make the meaning of "Work time "clearer, we changed it to" Work tenures ".

  1. Please revise the mentions to safety participation and conscientiousness in tables 2 and 3.

Response: Thank you for your kind advice. We have modified the mentions to safety participation and conscientiousness in tables 2 and 3.

  1. There is no explanation of the data analysis procedure in the method section, therefore it comes as a surprise that the results are presented for both Safety behavior and the two safety behavior subscales. Figures 2 and 3 are much like figure 1, they can actually be deleted, and a comment added to explain that the three figures are similar.

Response: Thank you for your kind advice. Since safety behavior are divided into safety compliance and safety participation for study, we delete the results of safety behaviors and Figure 1 of safety behaviors in section 4.4 in the results section. And we retained both the safety compliance and safety participation images because the trends are similar, but they represent different variables.

  1. DISCUSSION.
  2. A lot new information has been added to the discussion which should have first been introduced in the theory section. For instance, fairness theory, reciprocity, high power distance… These concepts should be introduced earlier on, and then discussed in the discussion with reference and comparison with similar empirical studies.

Response: Thank you for your kind advice. We added explanations of fairness and reciprocity in the introduction and revised them in the discussion. High power distance refers that there is a big difference between superior and subordinate in rank and status. Subordinates are more inclined to follow the willingness and arrangement of superiors and focus on tasks assigned by superiors. Since power distance is not a key factor in this study, we do not present its concept in the text. (See page 4-5)

The principle of equity in social exchange theory means that individuals must conduct internal cost-benefit analyses before participating in social exchange [30].

Social exchange theory holds that human interaction in organizations is essentially a series of exchanges based on the principle of reciprocity [31]. Reciprocity refers to the principle that when one party provides the resource that the other party needs, the other party will reciprocate with the resource that the other party needs. And negative reciprocity means that when one party's resources are damaged, it will retaliate to achieve balance. Abusive supervision will lead to negative reciprocity, which reduces the safety motivation of employees and reduces their investment in organizational safety construction, thereby showing a lower level of safety participation.

  1. In my view, the elaboration on the different types of motivation according to Ryan and Deci in the introduction and in the discussion is inappropriate as the study has not empirically distinguished or measured autonomous vs. controlled motivation.

Response: Thank you for your kind advice. We modified and deleted the elaboration on the theory of self-determination and different types of motivation. (See page 4)

Safety motivation refers to the willingness of employees to perform work in a safe manner, including their voluntary efforts to complete work tasks safely [24]. Organizational situational factors (leadership style) are related to the stimulation of employee‘s safety motivation [16,17]. Different leadership styles have different influences on safety motivation. Positive leadership styles can stimulate employees' safety motivation [25], and negative leadership styles will reduce employees’ safety motivations and increase occupational safety risks [26]. Abusive supervision, as a negative leadership style, may reduce employee‘s safety motivation.

  1. The results on full vs. partial mediation of motivation for the different safety behaviors should be further elaborated and discussed.

Response: Thank you for your kind advice. We further elaborated and discussed the mediation of motivation for the different safety behaviors. (See page 16)

This may be because safety compliance is a relatively passive behaviours, which is not significantly affected by many other factors. Therefore, safety motivation can full mediate the relationship between abusive management and safety compliance. However, safety participation is a relatively active behaviours, which requires more efforts and is affected by many other factors. Therefore, safety motivation only partially mediates the relationship between abusive management and safety participation.

  1. It is confusing to mix results on both “overall safety behaviors”, and on “specific safety behaviors”, I suggest starting with an overall comment on safety behavior, but discussing results only on specific safety behaviors (compliance and participation).

Response: Thank you for your kind advice. Since safety behavior is composed of safety compliance and safety participation, and this study studied safety compliance and safety participation respectively, we modified the presented results and specifically discussed safety compliance and safety participation when discussing the results.

  1. The authors should elaborate and provide richer inputs in the discussion; at the moment it is pretty much a repetition of the introduction and results.

Response: Thank you for your kind advice. We have adjusted the discussion. ( See page 15-16)

5.1.1 Abusive supervision and safety behaviour

Studies have shown that abusive supervision behaviours have a negative relationship with employees’ safety behaviours (including safety compliance and safety participation) [10]. Hypothesis 1 is verified in this study. The reason why employees in the petrochemical industry choose to lower their safety behaviour level after experiencing abusive supervision is due to the phenomenon of social exchange. According to the principle of negative reciprocity in social exchange theory, leaders conduct abusive supervision on employees, which makes employees realize which they get worthless rewards from leaders for their work, even insult and ridicule are not conducive to employees themselves. Based on the principle of negative reciprocity, employees regard the safety of the organization with this negative attitude and a lower level of safety compliance. Previous studies have pointed out that the relationship between Chinese managers and employees is characterized by “high power distance” [42], and traditional Chinese culture emphasizes the idea of peace and harmony as most valuable [43,44]. Therefore, employees will choose alternative retaliatory which under abusive supervision. That is, they will negatively treat organizational safety, be unwilling to participate in organizational safety construction, and show a lower level of safety participation.

5.1.2 The mediating role of safety motivation

The research found that safety motivation plays a mediating role in the relationship between abusive supervision and employee safety behaviours (safety compliance and safety participation). Hypothesis 2 is verified.

Previous studies have found that a leader with an active leadership style helps employees incorporate various company values, such as safety, independent work motivation, a positive work attitude, and better work performance [45]. In addition, this study found that when leaders improperly supervise employees, employees’ motivation for work is damaged, leading to their weaker motivation to perform safe behaviours. According to the theory of social exchange, when employees experience abusive supervision, they will take measures to respond similarly to the abusive supervisor. This is based on the principles of fairness and reciprocity in social exchange theory [31]. Therefore, after experiencing abusive supervision, employees will adopt negative attitudes and behaviours or reduce their own positive attitudes and behaviours, that is, reduce safety motivation.

Studies have found that there is a positive correlation between safety motivation and safety behaviours [46], which is consistent with the results of this research. The reason might be that an individual’s behaviour is driven by their motivation. Motivation is more helpful in stimulating desired behaviours and is associated with many positive results, such as promoting employees’ job performance [47,48]. Therefore, when employees have a higher level of safety motivation, they will be effectively motivated to comply with the safety regulations, show a higher level of safety compliance, and participate in safety construction.

Abusive supervision reduces the safety motivation of employees, which further reduces employee safety behaviours (safety compliance and safety participation). The reason why abusive supervision leads to the declination of safety motivation and employee safety behaviours may be that the personalized care and supportive leadership style of leaders can positively affect safety motivation [49,50], but abusive supervision does not provide personalized care and support, and is negatively correlated with supportive feelings such as belonging [10]. So, it negatively affects safety motivation and further leads to a reduction in the level of employee safety behaviour. In addition, this study found heterogeneity in the mediating mechanism of safety motivation in the relationship between different safety behaviours and abusive supervision. Safety motivation plays a complete mediating role in the relationship between abusive supervision and safety compliance and plays a partial mediating role in the relationship between abusive supervision and safety participation. These results imply that abusive supervision has different effects on different safety behaviours of employees. This may be because safety compliance is a relatively passive behaviours, which is not significantly affected by many other factors. Therefore, safety motivation can completely mediate the relationship between abusive management and safety compliance. However, safety participation is a relatively active behaviours, which requires more efforts and is affected by many other factors. Therefore, safety motivation only partially mediates the relationship between abusive management and safety participation.

11.Formal aspects: References are not in alphabetical order. Punctuation signs need to be reviewed. English needs to be reviewed by a native-level speaker.

Response: Thank you for your kind advice. The order of references has been modified, and punctuation has been reviewed. English has been reviewed by two native-level speakers.

Reviewer 3 Report

the study was well-done and well-written.

One Minor writing concern/suggestion

Figure 1 would be clearer if the two dependent variables (safety compliance and safety participation) were shown as separate variables. This would make research model clearer and align with the hypotheses)

Author Response

Thank you for taking time to provide this in-depth review of our paper. This valuable information has allowed us to improve the manuscript. Below, we clarify the changes that have been made.

1.Figure 1 would be clearer if the two dependent variables (safety compliance and safety participation) were shown as separate variables. This would make research model clearer and align with the hypotheses)

Response: Thank you for your kind advice. We have modified the model diagram as follows: (See page 3)

Round 2

Reviewer 2 Report

Dear authors,

Thank you for the consideration of previous comments. There are still issues to work on before publication.

Introduction: Cold violence should be accompanied by a definition. The sentence starting with "Studies have shown for pilots" is incomplete when it comes to the relationship between petro-chemical industry and unsafety as it is unrelated to abusive supervision. Throughout the text it is stated various times that Motivation is an attitude, please refer to classical definitions of motivation where it is referred to as an internal state, impulse but not an attitude and revise the references to motivation as an attitude (also in section 2.3)

The second part of this sentence "This mediating mechanism reflects the important role of safety motivation between leadership style and employee safety behaviour, and helps researchers to understand an indirect path such as abusive supervision that affects employee safety behaviour" needs reframing. 

When it comes to this sentence "Third, it further explored the moderating role of conscientiousness in the relationship between abusive supervision and employee safety" the moderating role is between motivation and safety.

Thank you for your new description of the sample employees, now the context for research is much clearer and interesting.

Regarding 3.2.4 please add the initial statement of the conscientiousness scale. 

Section 4.1. "there are six factors emerged" please correct grammar.

Section 4.2. There is no explanation for different factor models tested. You would be better off adding a Data analysis section in the method section.

Table 3. The interaction term between abusive supervision and conscientiousness is not clear.

The sentence starting According to Ribeiro would fit better in a data analysis section.

There is no table 4, the manuscript currently presents table 3 and 5. THe results described as in table 5 are now in table 3.

The results about different conditions of conscientiousness do not match between the text description and currently table 5 for example in the text it is stated that CI (-0.106, -0.106). Review also if significance - nonsignificance is appropriately interpreted according to the data portrayed in the table. Check all results, table and results descriptions.

The authors describe results for three conditions of conscientiousness but depict only two conditions in the figures. Besides, the figures depict responsibility instead of conscientiousness.

DISCUSSION. Maybe hypotheses are empirically supported rather than verified. 

5.1.1. the reason why employees choose to lower their safety behaviour "may be" related or explained through social exchange (rather than "is", as social exchange per se has not been tested in this study).  The new sentences including the words insult and ridicule do not make sense, I feel it is due to grammatical errors. Overall this 5.1.1. paragraph needs a calm elaboration.

There are two sections numbered 5.1.2. The first one repeats section 5.1.1 and should be deleted.

Second section 5.1.2. In my view, it is more appropriate to start with "leaders improperly supervising employees" than with the first current sentence which is a bit vague about active? or positive? leadership style (which again mentions attitudes not tested in this study. This mentions to positive leadership are repeated in the paragraph starting with "abusive supervision" it is a bit confusing. 

Section 5.1.2. The ideas in this sentence "This may be because safety compliance is a relatively passive behaviours, which is not significantly affected by many other factors. Therefore, safety motivation can completely mediate the relationship between abusive management and safety compliance. However, safety participation is a relatively active behaviours, which requires more efforts and is affected by many other factors. Therefore, safety motivation only partially mediates the relationship between abusive management and safetyparticipation." The authors reflect on active vs. passive behaviours. I suggest instead on in-role (compliance) or extra-role behaviours (participation) may better explain the influence of the mechanisms found in the results.

Section 5.13. The first sentence about moderating the mediating role needs a revision of grammar. The second paragraph in this section develops arguments building on stability of the relationship, which have not been tested or introduced in the manuscript til this section. I suggest the authors stick to personality of employees which makes them more responsible in complying with safety demands regardless of the influence of the supervisor instead of introducing a new factor in this part of the manuscript.

Section 6. THe authors suggest for future research to deal with issues such as team atmosphere and workplace gossip. Instead I suggest incorporating more directly related issues to the current manuscript such as safety culture, or voice behaviours within the team.

Hope these comments help you improve your manuscript.